# A humanized mouse model for adeno-associated viral gene therapy

Mercedes Barzi[1,14], Tong Chen[1,2,14], Trevor J. Gonzalez[2], Francis P. Pankowicz[3], Seh Hoon Oh[4], Helen L. Streff [5], Alan Rosales [5], Yunhan Ma[1], Sabrina Collias[1], Sarah E. Woodfield[6,7], Anna Mae Diehl [4], Sanjeev A. Vasudevan[6,7], Thao N. Galvan [7,8], John Goss[7,8], Charles A. Gersbach [5,9,10,11,12], Beatrice Bissig-Choisat[1], Aravind Asokan [2,5,10,11,12] & Karl-Dimiter Bissig [1,4,5,9,11,12,13] ✉

Clinical translation of AAV-mediated gene therapy requires preclinical development across different experimental models, often confounded by variable transduction efficiency. Here, we describe a human liver chimeric transgene-free *Il2rg⁻/⁻/Rag2⁻/⁻/Fah⁻/⁻/Aavr⁻/⁻* (TIRFA) mouse model overcoming this translational roadblock, by combining liver humanization with AAV receptor (AAVR) ablation, rendering murine cells impermissive to AAV transduction. Using human liver chimeric TIRFA mice, we demonstrate increased transduction of clinically used AAV serotypes in primary human hepatocytes compared to humanized mice with wild-type AAVR. Further, we demonstrate AAV transduction in human teratoma-derived primary cells and liver cancer tissue, displaying the versatility of the humanized TIRFA mouse. From a mechanistic perspective, our results support the notion that AAVR functions as both an entry receptor and an intracellular receptor essential for transduction. The TIRFA mouse should allow prediction of AAV gene transfer efficiency and the study of AAV vector biology in a preclinical human setting.

Adeno-associated virus (AAV) is a commonly used gene therapy vector, and after many years of vector development and clinical trials, two different AAV hemophilia gene therapies (Hemgenix and Roctavian) have been approved by FDA[1,2]. An important lesson from preclinical development was that experimental animal models poorly predict human AAV transduction and in vivo pharmacology. Albeit all currently evaluated AAV serotypes display hepatotropism in experimental animal models, there are differences in uptake, nuclear entry, uncoating, second-strand synthesis, and transgene expression/silencing in the liver and other organs[3,4]. While non-human primates may only be predictive for human hepatocyte transduction for certain serotypes[3], ethical and cost constraints limit these models. Human

¹Alice and Y. T. Chen Center for Genetics and Genomics, Division of Medical Genetics, Department of Pediatrics, Duke University Medical Center, Durham, NC 27710, USA. ²Department of Molecular Genetics and Microbiology, Duke University Medical Center, Durham, NC 27710, USA. ³Center for Cell and Gene Therapy, Stem Cells and Regenerative Medicine Center, Baylor College of Medicine, Houston, TX 77030, USA. ⁴Department of Medicine, Division of Gastroenterology, Duke University Medical Center, Durham, NC 27710, USA. ⁵Department of Biomedical Engineering, Duke University Pratt School of Engineering, Duke University, Durham, NC, USA. ⁶Michael E. DeBakey Department of Surgery, Divisions of Pediatric Surgery and Surgical Research, Baylor College of Medicine, Houston, TX 77030, USA. ⁷Department of Surgery, Texas Children's Hospital, Houston, TX 77030, USA. ⁸Michael E. DeBakey Department of Surgery, Division of Abdominal Transplantation and Division of Hepatobiliary Surgery, Baylor College of Medicine, Houston, TX 77030, USA. ⁹Duke Cancer Center, Duke University Medical Center, Durham, NC 27710, USA. ¹⁰Department of Surgery, Duke University Medical Center, Durham, NC 27710, USA. ¹¹Duke Regeneration Center, School of Medicine, Duke University, Durham, NC, USA. ¹²Center for Advanced Genomic Technologies, Duke University, Durham, NC, USA. ¹³Department of Pharmacology and Cancer Biology, Duke University Medical Center, Durham, NC 27710, USA. ¹⁴These authors contributed equally: Mercedes Barzi, Tong Chen. ✉e-mail: karldimiter.bissig@duke.edu

liver chimeric mice are a step forward in this regard and allow validation of AAV in primary human hepatocytes in vivo[5–9]. These humanized mice achieve high human chimerism (>60% human)[10,11] and can be used for evaluation of AAV transduction efficiencies of human hepatocytes in vivo[12–14]. Unfortunately, many AAV serotypes validated for transduction efficiency in humanized mice transduce murine hepatocytes much better than human hepatocytes[12,13]. Also clinically used serotypes, such as AAV8 and AAV9, seem to transduce murine hepatocytes much more efficiently than human hepatocytes[12,14]. This "sink" effect arising from uptake into murine hepatocytes confounds findings obtained from human liver chimeric mice utilized for validation of AAV gene therapy vectors. Thus, there is a critical unmet need to develop an improved humanized mouse model that allows specific and selective interrogation of AAV in human hepatocytes without background effects imposed by murine hepatocytes or other extrahepatic tissues.

Recently, a universal AAV receptor (AAVR) has been identified[15]. This work showed that murine AAVR deficient cells in $Aavr^{-/-}$ mice were highly resistant to AAV infection with several serotypes. In the present study, we combine murine AAVR deficiency with human liver chimerism and demonstrate an increased and preferential AAV transduction efficiency of human hepatocytes. We use this novel humanized mouse model also to demonstrate utility for other tissues and clinically relevant applications.

## Results

### Generation of the TIRFA strain

In order to validate AAV gene therapy in human hepatocytes, we deleted the *Aavr* gene using a CRISPR approach in zygotes of TIRF mice[16] to generate a the transgene-free $Il2rg^{-/-}/Rag2^{-/-}/Fah^{-/-}/Aavr^{-/-}$ (TIRFA) mouse strain (Fig. 1a and b). We previously showed that TIRF mice can be repopulated with human hepatocytes similarly to the first-generation humanized mouse models[5–9] but without any potential effects originating from the transgenes[16]. Homozygous TIRFA mice do not have any obvious pathology and can be transplanted with human hepatocytes as we have described previously[6,16,17]. Humanized TIRFA mice secrete human albumin into the blood to a similar degree as control humanized TIRF ($Aavr^{+/+}$) mice (Fig. 1c), displaying increased concentration over time with increased humanization (Fig. 1d). The typical histological morphology of human hepatocytes in chimeric mice could be observed in humanized TIRFA mouse (Fig. 1e).

### AAV tropism of human liver chimeric TIRFA mice.
Having established human liver chimeric TIRFA mice, we next validated the tropism of AAV8, a serotype known to enter via AAVR[15], using an AAV backbone expressing the reporter tandem dimer Tomato (tdTomato) from the ubiquitously expressed chicken β-actin (CBA) promoter. We injected AAV8 intravenously ($1 \times 10^{12}$ gc/mouse) into humanized mice (TIRF and TIRFA) and analyzed transduction efficiencies one month later. As expected, we observed minimal transduction in most organs (kidney, spleen, brain, lung, skeletal muscle) when the murine host was AAVR-deficient (Fig.1f). The notable exceptions were the heart and the liver. Sporadic cardiac expression appears to suggest that alternative mechanisms leading to transduction may be in play, potentially enabled by molecules such as GPR108[18]. More importantly, in contrast to conventional human liver chimeric mice (with murine AAVR such as TIRF), the humanized TIRFA mouse did exclusively express the reporter transgene in human and not in murine hepatocytes (Fig. 1g). In order to confirm and quantify these observations, we humanized TIRFA mice (*n* = 10) from two different hepatocyte donors and injected the same AAV tdTomato reporter construct packaged in AAV8 or AAV9 capsids ($1 \times 10^{12}$ gc/mouse, i.v.). Our results confirmed the human hepatotropism for both AAV serotypes in TIRFA mice, whereas in control humanized TIRF mice, the majority of AAVs transduced the murine hepatocytes (Fig. 2a, c). Quantification revealed that significantly more human hepatocytes are transduced in TIRFA compared

to TIRF mice when injected with the same AAV at the same dose (Fig. 2b, d). More recently, capsids with high human hepatotropism have been described[13,19]. We used such a capsid, NP-59[19], and injected again at the same dose and route humanized TIRF and TIRFA mice. We could confirm human hepatotropism in both humanized TIRF and TIRFA, but the capsid retained the ability to transduce murine hepatocytes, particularly in non-humanized mice (Supplementary Fig. 1).

To improve our mechanistic understanding of AAVR deficiency in the context of AAV-injected TIRFA mice, we performed in situ hybridization for the detection of AAV DNA and RNA in the livers. When analyzing AAV8 and AAV9 injected humanized TIRFA mice, we could detect abundant AAV DNA and RNA in human hepatocytes (Fig. 3). Interestingly, we could detect AAV DNA (Fig. 3a, c, e) not only in humans but also in murine hepatocytes of TIRFA mice, however, accumulation in murine cells was less abundant in TIRFA compared to TIRF mice. We could also detect a strong signal for AAV RNA in human cells (Fig. 3b, d, f), whereas in TIRFA mice, there was only a very weak or absent signal for murine cells.

Next, we measured AAV genome copies in the liver and other organs (kidney, lung, muscle, spleen, and heart) (Supplementary Figs. 2 and 3) and could detect >99% of AAV in the chimeric livers, regardless of AAV serotype. Albeit based on low copy number levels, some differences were observed in AAV8 vs AAV9 treated animals, particularly for peripheral tissues. Notably, while AAV8 showed minimal variation in transduction across extra-hepatic tissues, AAV9 showed decreased transduction in spleen, muscle, etc., suggesting that AAVR may have a differential impact on tissue transaction of different serotypes.

**Other applications of humanized TIRFA mice.** To fully leverage humanization in TIRFA mice, we generated a human teratoma model, which produces human tissue from all three germ layers. Such a model should enable validation of AAV tropisms in any desired human tissue. For this purpose, we injected subcutaneously human induced pluripotent stem cells into TIRFA mice and let teratoma develop over 3–4 months. Mice were then injected intravenously with AAV9 ($1 \times 10^{12}$ gc/mouse) expressing GFP from the CBA promoter. Four weeks after injection, we analyzed teratoma for transduction with the AAV reporter and co-localization with known tissue markers (Fig. 4a–c). Similarly, we transplanted patient-derived human liver cancer tissue into humanized TIRFA livers and injected a AAV9-tdTomato vector at the same dose. As expected, we detected tdTomato in the primary human hepatocytes but also in the human liver cancer sample (Fig. 4d, e).

## Discussion

Over the past decade, the liver has emerged as a promising target organ for expressing therapeutic transgenes utilizing AAV vector-mediated delivery. Ongoing liver-directed gene therapy clinical trials are focused on Pompe disease, Urea cycle disorders such as ornithine transcarbamylase (OTC) deficiency, methylmalonic acidemia (MMA), and Wilson disease, among others, while first AAV gene therapies for hemophilia A and B have already been approved for use in patients[1,2]. The increasing number of liver-directed AAV gene therapies parallels the rapid emergence of engineered recombinant AAV capsids[20]. These capsids are developed to evade neutralizing antibodies and/or to increase a particular tissue specificity. However, since AAV capsids display species-specific transduction efficiencies[3,4], engineered capsids will benefit from being evaluated in a human setting before clinical use.

Here, we introduce a new experimental tool, a human liver chimeric mouse deficient of the murine AAVR. Almost all natural variants and engineered capsids require AAVR for transduction in mice[15,18]. Hence, it is not surprising that transduction of human hepatocytes increases significantly (2–7-fold) in humanized mice without AAVR (TIRFA) compared to mice with a functional receptor (TIRF). Interestingly, we could detect transgenic vector genomic DNA shortly after

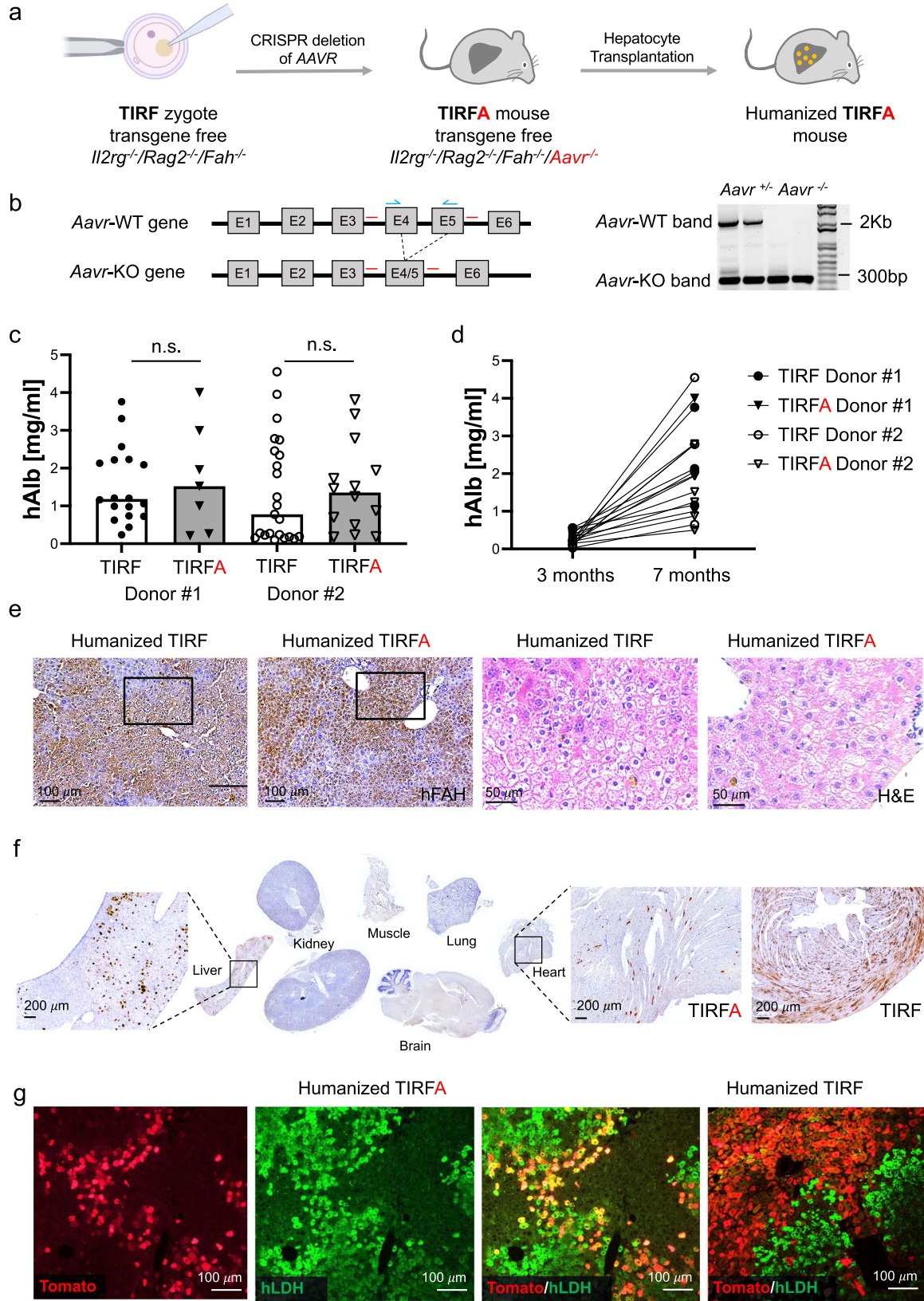

the injection of AAV vectors into humanized TIRFA, not only in human but also in murine cells. There was also a tendency for less vector DNA in murine cells of TIRFA than in TIRF mice. Even more pronounced was the difference at the RNA level: while murine hepatocytes in TIRF mice had a clear and strong signal for both the AAV8 and AAV9 transgene, there was, at best, a faint signal in a few murine hepatocytes of TIRFA

mice. In situ hybridization is semiquantitative, and the assessment of transgenic DNA and RNA levels in individual liver cells is challenging. Nevertheless, our results support the notion that AAVR likely functions as an intracellular receptor or host factor essential for transduction.

Increased transduction of human hepatocytes in humanized TIRFA mice will allow studying many biological questions, such as AAV

**Fig. 1 | Generation of the TIRFA strain. a** AAVR gene was knocked out in TIRF zygotes using CRISPR/Cas9 gene editing tools. **b** AAVR gene scheme showing the sgRNAs used to target exons 4 and exons 5 and the primers binding sites used to screen the mice. On the right, DNA gel electrophoresis shows PCR bands of heterozygous and homozygous mice. **c, d** Human albumin levels of humanized TIRF control and TIRFA mice transplanted with two different hepatocyte donors ($n=18$ for TIRF donor#1, $n=23$ for TIRF donor#2, $n=7$ for TIRFA donor#1 and $n=14$ for TIRFA donor#2) **c** and its increase over time **d** ($n=3$ for TIRF donor#1, $n=4$ for TIRF donor#2, $n=3$ for TIRFA donor#1 and $n=6$ for TIRFA donor#2), **e** FAH immunostaining and H&E of TIRF and TIRFA humanized mice livers **f** tdTomato immunostaining of different organs from a TIRFA mouse injected with AAV8-tdTomato virus.

On the right, amplification of the heart tissue. **g** Immunofluorescence staining of the livers from TIRF and TIRFA humanized mice injected with AAV8 virus expressing tdTomato (red). Human cells are stained with hLDH (green). Results observed in (**b**) and (**e–g**) were observed in at least 3 independent samples per condition. sgRNA single guide RNA, PCR polymerase chain reaction, H&E Hematoxylin and Eosin, FAH fumarylacetoacetate hydrolase, AAV adeno-associated virus, RFP red fluorescent protein, hLDH human lactate dehydrogenase. Data are presented as means ± SD. Significance was validated with a two-sided student t-test. Source data are provided as a Source Data file. Scale bars represent 50, 100, or 200 μm, respectively.

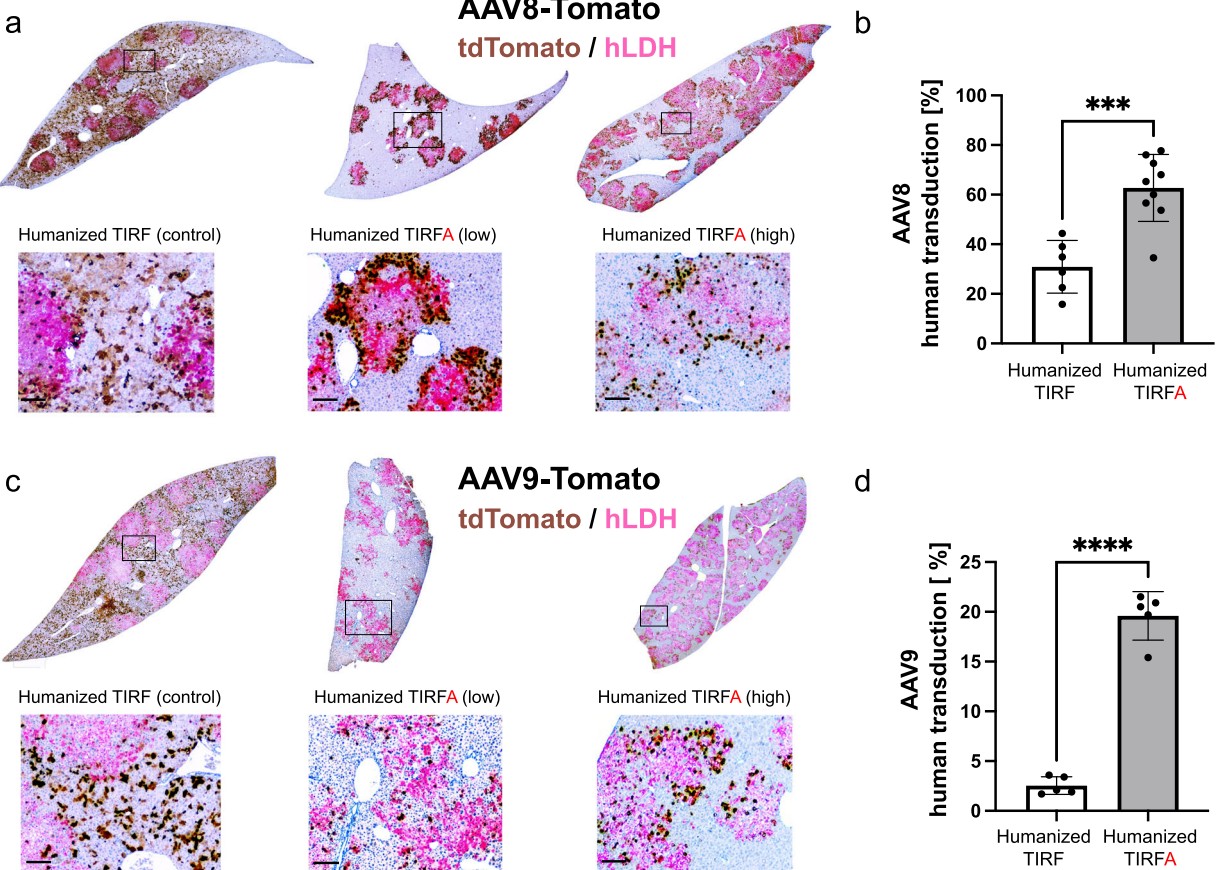

**Fig. 2 | Tropism of AAV viruses in humanized TIRFA mice.** Humanized TIRF and TIRFA mice were injected ($1 \times 10^{12}$ gc/mouse) with AAV8 (**a, b**) or AAV9 (**c, d**) expressing tdTomato and harvested after a month. Immunohistochemistry showing co-staining of hLDH (magenta) and tdTomato (brown) in livers of humanized TIRF and TIRFA mice (low and high repopulated). Quantification of human hepatocytes expressing tdTomato from mice injected with (**b**) AAV8 ($n=6$ and $n=9$

lobes for humanized TIRF and TIRFA, respectively) or (**d**) AAV9 ($n=5$ lobes for both humanized TIRF and TIRFA). hLDH human lactate dehydrogenase. Data are presented as means ± SD. Significance was validated with a two-sided student t-test. ***$P \le 0.001$ and ****$P \le 0.0001$. Source data are provided as a Source Data file. Scale bars represent 100 μm.

vector genome silencing and/or persistence of transgene expression in human target cells. Notably, recent studies have shown that AAV-mediated transgene expression in monkey hepatocytes is bi-phasic – with episomal genomes enabling high but transient expression and integrated genomes resulting in lower but stable expression[21]. A better understanding of these phenomena in human hepatocytes is critical. Moreover, these translational questions have become more important with the FDA approval of the first AAV gene therapies in the liver, and the TIRFA mouse offers a robust preclinical platform to study some of these important clinical challenges. Humanized mice with wildtype AAVR (TIRF, FRG, uPA, or TK-NOG strains) suffer from the expression of the AAV transgene in murine hepatocytes, and depending on the application, this can be limiting. In the previously mentioned example

of gene silencing, transduced murine cells might continue transgene expression while human cells are silenced and thereby confound the interpretation of the analysis. In an alternative scenario, several bioengineered AAV capsids (LK03, NP-59, etc.) with preferential transduction in human hepatocytes have been developed. These AAV capsids (amongst others) have been shown to influence the epigenetic marking of episomal genomes in a species-dependent manner[22]. These latter AAVs have been selected for increased human hepatocyte tropism in humanized FRG (AAVR wildtype) mice. As expected, they have an improved human tropism and transduce human hepatocytes equally well in TIRF and TIRFA mice. However, they retain the capacity to transduce murine hepatocytes in humanized and particularly non-humanized TIRF (AAVR wildtype) mice. Thus, the TIRFA mouse offers a

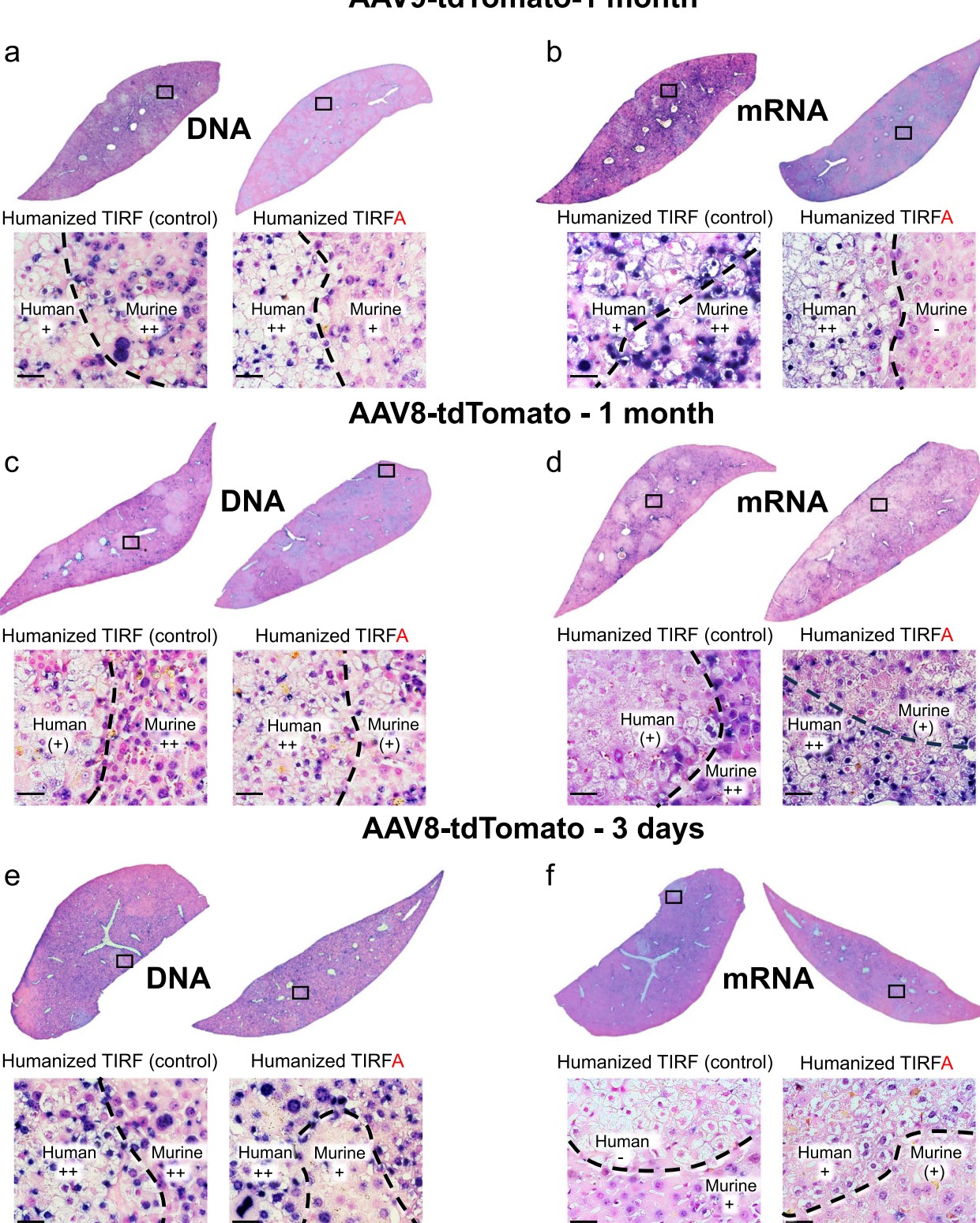

**Fig. 3 | Detection of AAV DNA and RNA in human liver chimeric mice.** Humanized TIRF and TIRFA mice were intravenous injected with $1 \times 10^{12}$ gc/mouse AAV8-tdTomato or AAV9-tdTomato. Shown are representative in situ hybridization of AAV DNA (**a**, **c**, **e**) or RNA (**b**, **d**, **f**) from chimeric liver sections. Detection of DNA (**a**) and RNA (**b**) of AAV9-tdTomato one month after injection. Detection of DNA (**c**) and RNA (**d**) of AAV8-tdTomato one month (**c**, **d**) and 3 days (**e**, **f**) after injection. Murine with adjacent human areas is shown and semiquantitatively scored: "−" no nucleotide detection, "(+)" minimal, "+" little, "++" abundant DNA, respectively RNA. The boxed area is shown with higher magnification. Results observed in these figures were observed in at least 3 independent samples per condition. Scale bars represent 25 µm.

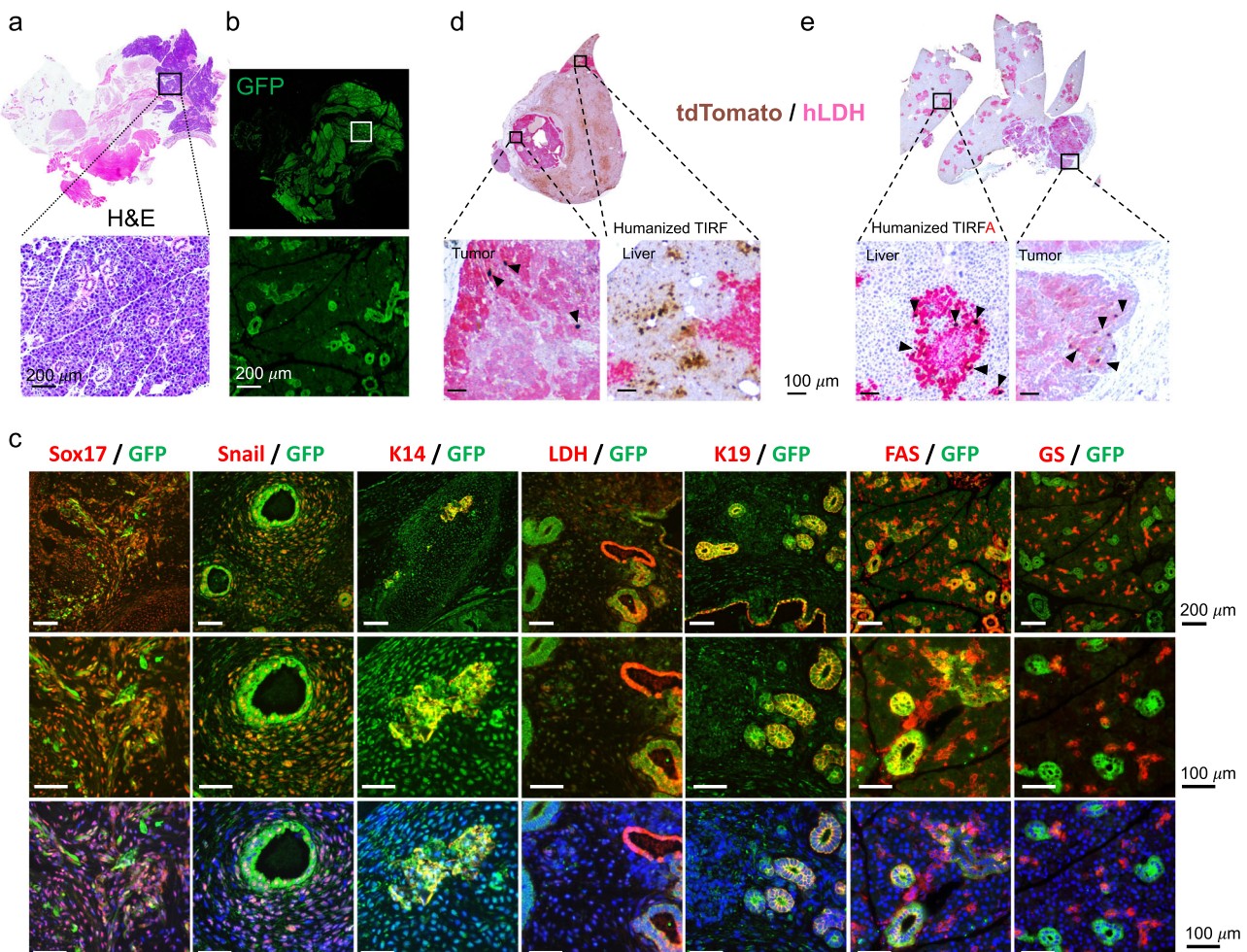

**Fig. 4 | Tropism of AAV viruses for different human tissues in humanized TIRFA mice.** H&E staining of a teratoma from a TIRFA humanized mice (**a**) injected with AAV9-GFP virus (**a, b, c**) teratoma co-stained with GFP (green) and different markers (red) (**c**): Sox17, Snail, Cytokeratin 14 (K14), lactate dehydrogenase (LDH), cytokeratin 19 (K19), FAS receptor and Glutamine synthetase (GS). (**d, e**) Human tumor of a TIRF (**d**) and TIRFA (**e**) humanized mouse injected with AAV9-tdTomato stained with hLDH (magenta) and tdTomato (brown). H&E Hematoxylin and Eosin. GFP green fluorescent protein, RFP red fluorescent protein, hLDH human lactate dehydrogenase. Representative results were observed in at least 3 independent samples per condition. Scale bars represent 100 or 200 μm, respectively.

unique opportunity to understand AAV capsid-genome interactions and human-specific mechanisms of transgene expression and silencing.

AAV gene therapy is also used for many other target organs but since the majority of natural serotypes are home to the liver, there are a lot of efforts for de-targeting the liver and decreasing the natural hepatotropism[23]. To generate a proof-of-concept that TIRFA mice could also be used for extrahepatic applications, we performed a feasibility study by generating teratomas in TIRFA mice. Teratomas form tissues and primary cells from all three germ layers and are therefore often used as a functional test for pluripotency of stem cells[24]. Here we used the teratoma assay to generate human tissue and investigate if systemically injected AAV would be able to transduce stem-cell-derived primary human tissue. As far as we know, there are no such studies published but we think that the teratoma assay in the context of the TIRFA mouse is a convenient way to study human cellular tropism of AAV serotypes. The advantage of this method is that no xenografting expertise is required and that several human tissues are present in the same teratoma. The drawback is that teratomas develop stochastically and each teratoma forms different numbers of different tissues. Hence comparing teratomas to each other or, in our case, quantifying transduction efficiencies of AAV is almost impossible. Nevertheless, the teratoma assay in TIRFA mice is a good first step to

evaluate if AAV is able to transduce the desired human primary cell. Similarly, we think that quantification of primary xenograft cancer models is problematic since they are polyclonal and depend on sampling of the heterogenous human cancer tissue. To mitigate this quantification problem, cancer cell lines can be used, or for primary tissue, proper xenograft transplantation models can be applied.

In summary, we introduce a versatile mouse model that allows the interrogation of AAV tropism and biology selectively in human cells in vivo. We demonstrate that AAV vectors, when dosed intravenously in human liver chimeric TIRFA mice, can more readily transduce human rather than murine hepatocytes and can be used for the assessment of other human cells and target tissues. The TIRFA mouse overcomes significant limitations posed by current experimental animal models and should allow the prediction of in vivo pharmacology of liver-directed AAV gene therapies as well as studying AAV biology in a human setting.

## Methods

### Mouse maintenance, experiments, and euthanasia
All aminals were maintained under a standard 12-h dark/light cycle with water and chow (Picolab Rodent diet 20, LabDiet, cat# 5053) provided ad libitum. All animal experiments were approved by the Duke University Institutional Animal Care and Use Committee. Euthanasia was

done via carbon dioxide overdose followed by decapitation or bilateral opening of the thoracic cavity to ensure immediate death.

## Generation of TIRFA mice by CRISPR/Cas9 gene editing in zygotes

Design and cloning of sgRNA was done as previously described[25]. In brief, complementary oligonucleotides were cloned into the DR274 vector (Addgene, cat# 42250).

We targeted exons 4 and 5 of the mouse *Aavr* gene (Fig. 1a) using the following sgRNAs:

*Aavr*-sgRNA Ex4: 5′ ACTTGCGGAGTACCAGATACAGG 3′
*Aavr*-sgRNA Ex5: 5′ CATTCTTAGGCAGGGTGATCTGG 3′

A T7 bacterial promoter sequence was inserted into the pX330-U6-Chimeric_BBCBh-hSpCas9 vector (Addgene plasmid # 42230) upstream of the Cas9 transcription start site. DR274 vectors were cut using DraI (NEB) and pXX30–spCas9 was digested with NcoI and NotI. Vectors were gel purified using the Zymoclean Gel DNA Recovery Kit (Zymo, Cat#11–301).

sgRNAs and Cas9 were in vitro transcribed in vitro using MEGA-shortscript T7 Transcription Kit (Life Tech, AM1354) and mMessage mMachine T7 ULTRA Kit (Life Tech AM1345), respectively. The resulting RNAs were purified using RNA Clean & Concentrate columns (Zymogen, cat# R1015) and eluted in RNAse-free water, and synthesis was verified by polyacrylamide gel electrophoresis.

A mix of 15 ng/μL of each sgRNA and 60 ng/μL of Cas9 mRNA in PBS was used for the zygote's microinjection. SpCas9 and sgRNA were injected into homozygous zygotes from TIRF (Transgene free *Il2rg*$^{-/-}$/*Rag2*$^{-/-}$/*Fah*$^{-/-}$) mice.

We obtained nineteen founder pups, four of which died after birth. All of them were screened for deletions. Two of the dead pups were wild-type, and the other two were homozygous. Seven out of the fifteen pups that survived were heterozygotes for an *Aavr* deletion. We obtained a total of four deletions, some of them shared by more than one mouse, and all of them led to a premature stop codon after the junction site. Mice with confirmed deletion of *Aavr gene* were back-crossed twice with the TIRF strain to eliminate any possible off-target mutations. Heterozygote F2 mice were crossed with each other to obtain homozygote pups. As previously described, homozygous males could not breed[26]. To generate homozygous animals used in this paper, we crossed heterozygous males with homozygous females.

## Genotyping of the TIRFA strain

All of the pups obtained were screened by PCR using the following primers:

*Aavr*_Ex4 For: 5′-ACAGTTGCCGGTTCCTTCAC-3′
*Aavr*_In5 Rev: 5′-CCACATGCACATCACAACCTC-3′

The expected PCR bands for wild-type and knocked-out mice were 2.3 Kb and 250 bp, respectively. PCR bands corresponding to deleted alleles were purified and sent for Sanger sequencing. Once the junction was defined, the further offspring were genotyped by Transnetyx Incorporation.

## Human hepatocyte isolation

We isolated human hepatocytes by the two-step collagenase perfusion method as we described previously[12]. In brief, we cannulated the largest portal veins with a silicon tubing system connected to a peristaltic pump, then flushed the liver with ice-cold basic perfusion solution (BPS: 10 mM Hepes buffer) followed by perfusion with BPS containing 0.5 mM EGTA to prevent the formation of blood clots. The liver was then perfused with warm collagenase solutions (2 mg/dl collagenase) until the organ became soft. We cut the liver into small pieces (2–3 cm³) and released hepatocytes into the solution by applying minor shear stress (with forceps) on the pieces. Hepatocytes were immediately washed in ice-cold BPS containing 0.5% BSA, centrifuged (3 times 50×*g*, 5 min), and cryopreserved in NG5A (ChemQ

Bioscience.). The IRB of Baylor College of Medicine and/or Duke approved the procurement of human tissue. Informed consent was obtained where "human subject research" was determined.

## Transplantation of human hepatocytes into TIRF-AAVR-KO strain

Cryopreserved hepatocytes were transplanted into 8-week-old TIRFA mice as previously described[12,16,17]. In brief, the abdominal cavity was opened by a midabdominal incision, and 3x10⁶ human hepatocytes in a volume of 100 μl PBS were injected into the spleen. Immediately after transplantation, selection pressure towards transplanted human hepatocytes was applied by withdrawing the drug NTBC from the drinking water in the following steps: 2 days at 25%, then 2 days at 12%, and eventually 2 days at 6% of the colony maintenance dose (100% = 7.5 mg/l) prior to discontinuing the drug completely[10]. In order to determine the extent of human chimerism, we measured human albumin (ELISA, Bethyl laboratories) in the murine blood.

Both female and male animals were used for humanization.

## AAV production

A triple-plasmid transfection protocol was used to generate rAAV vectors[27]; the transfection mixture contained: (1) the pXR helper plasmid; (2) the adenoviral helper plasmid pXX6–80; and (3) the tandem dimer Tomato (tdTomato), driven by a CBA promoter, flanked by AAV2 ITRs or GFP driven by the CBA promoter. Vector purification was carried out using iodixanol gradient ultracentrifugation followed by desalting with ZebaSpin desalting columns (40 K MWCO; Thermo-Scientific, Waltham, MA, USA). vg titers were obtained by qPCR (LightCycler 480; Roche Applied Sciences, Pleasanton, CA, USA) using primers designed to selectively bind AAV2 ITRs:

For: 5′-AACATGCTACGCAGAGAGGGAGTGG-3′
Rev: 5′-CATGAGACAAGGAACCCCTAGTGATGGAG-3′

## Determination of vector genome biodistribution by quantitative PCR

DNA was extracted from tissue using a Purelink Genomic DNA Mini kit (Thermo Fisher Scientific) following manufactures instructions. Vector genomes were quantified via qPCR, using a plasmid standard and primers targeting the SV40 polyA in the AAV genome. LightCycler 480 SYBR Green I Master Mix (Roche) was used following manufacturer instructions for cycling conditions. The biodistribution of viral genomes is calculated as the ratio of vector genomes per microgram of DNA extracted and represented as fold change relative to mock.

Primers:
SV40_For: 5′-GCAGACATGATAAGATACATTGATGAGTT-3′
SV40_Rev: 5′-AGCAATAGCATCACAAATTTCACAA-3′

## Generation of teratoma and liver cancer xenograft models

Human induced pluripotent stem cells (iPSC) were cultured on matrigel (Corning) and in mTeSR media (Stemcell Technologies). In total, 2–5 million iPSC were scraped from culture plates and injected in 100 μl mTeSR subcutaneously to TIRFA mice. Over 3–4 months, human teratomas developed. Mice were injected with AAV (1 × 10¹² gc/mouse, i.v.) when teratomas were visible and reached 1 cm diameter.

Cryopreserved pieces of pediatric liver cancer (hepatoblastoma) were implanted onto the liver as previously described. In brief, the abdominal cavity of TIRFA mice was incised, and ~50 mm³ pieces were glued (Vetbond tissue adhesive, TM) onto the left lower liver lobe. The IRB of Baylor College of Medicine and Duke University Medical Center approved the procurement of human tissue. Informed consent was obtained from patients.

## Immunostaining chimeric livers and teratomas

Immunostaining was performed on formalin-fixed paraffin-embedded samples. Paraffin sections (5 μm thick) were dewaxed and rehydrated.

For immunohistochemistry (IHC), all the samples were treated with hydrogen peroxide and blocked using corresponding blocking serum as recommended by the manufacturer's instructions.

Rabbit anti-hFAH primary antibody (Sigma Aldrich, cat# SAB2108553) was diluted 1:250 and incubated overnight at 4 °C. IHC was developed following the rabbit kit (Vector Labs, cat# PK4001) manufacturer's instructions.

Co-staining of tdTomato (red for IF, brown for IHC) and hLDH (green for IF, magenta for IHC) required antigen retrieval with citrate buffer pH 6.0 for 30 min. After blocking with 2.5% horse serum, samples were incubated overnight at 4 °C with mouse monoclonal anti-hLDH antibody (Santa Cruz cat# LDH (H-10) sc-133123) in combination with rabbit anti-RFP antibody (Rockland, cat# 600–401–379), both diluted 1:100 in antibody diluent (Abcam, cat# ab642111) at 4 °C. Immunofluorescence (IF) was developed using fluorescent-labeled secondary antibodies (Jackson ImmunoResearch Laboratories) and IHC was developed following Vector ImmPRESS Duet immunohistochemistry kit manufacturer's instructions (Vector Labs, cat# MP-7714–15).

Co-staining of GFP (brown color) and LDH (magenta color) was performed in two steps. First, GFP immunostaining was performed by incubating the samples overnight at 4 °C with chicken anti-GFP (Abcam cat# ab13970) diluted 1:1,500 in antibody diluent. After washing, samples were incubated with anti-chicken biotinylated secondary antibody (Vector Labs, BA-9010) diluted 1:200 for 30 min, and the signal was developed using avidin/biotin/DAB system (Vector Labs, cat#PK4001). After developing, antigen retrieval with citrate buffer pH 6.0 for 30 min was performed, and samples were incubated overnight at 4 °C with anti-LDH antibody diluted 1:100. Magenta color was developed using Vector ImmPRESS Duet kit (Vector Labs, cat# MP-7714-15).

Immunofluorescence of teratoma samples was performed co-incubating overnight at 4 °C with chicken anti-GFP (Abcam cat# ab13970) diluted 1:200, in combination with the corresponding primary antibody diluted in abcam antibody diluent. The following antibodies required 15 min of antigen retrieval in citrate buffer pH 6.0: Rabbit anti-Sox17 (Millipore, cat# 09-038-1) diluted 1:150, rabbit anti-Snail (Abcam, cat# ab17732) diluted 1:500, mouse anti-LDH (Santa Cruz, cat# sc-133123) diluted 1:100 and anti-Rabbit Cytokeratin 14 (Abcam, cat# ab51054) diluted 1:1000. Rabbit anti-Cytokeratin 19 (Abcam, cat# ab52625) required 15 min EDTA pH 9.0 buffer antigen retrieval (Abcam, AB93684) and was used at a dilution of 1:400. Rabbit anti-GS (Abcam, cat# ab73593) was diluted 1:500 and rabbit anti-FAS (Santa Cruz, cat# sc-715) was diluted 1:50. After washing primary antibodies, samples were incubated with corresponding Alexa-fluor fluorescent-labeled secondary antibodies (Jackson ImmunoResearch Laboratories).

### Quantification of human cells transduced with tdTomato
Co-stained human cells (hLDH and RFP positive) and a total of human cells (hLDH positive only) from 5 different lobes were quantified using ImageJ software (https://imagej.nih.gov/ij/).

A minimum of 2000 cells were quantified, and the results are expressed as a percentage of human cells expressing tdTomato.

### In situ reverse transcriptase PCR (RT-PCR) for the detection of AAV mRNA
The protocol of in situ hybridization was modified from previous reports[28,29]. Paraffin sections (5 μm) were dewaxed, and antigen retrieval was performed using 10 mM sodium citrate buffer at 100 °C for 10 min. Subsequently, slides were given a DNase treatment with 2.5 units DNase (Thermo Scientific, EN0521) for 1 hr at 37 °C. The reaction was inactivated in 5 mM EDTA for 10 min. The slides were immersed in 2× SSC (Invitrogen, AM9763) for 5 min, followed by 95% 100% ethanol for 5 min each at room temperature and air-dried. For the reverse transcription (RT) reaction, 120 μl of RT reaction solution (12 μl 10 × RT buffer, 24 μl 25 mM MgCl₂, 12 μl 10 mM dNTP, 3 μl RNase Out, 6 μl Reverse transcriptase, 6 μl 100 nM reverse primer, 57 μl DEPC treated water) was added to each slide. The RT cycle ran at 30 °C for 10 min, 42 °C for 30 min, and 90 °C for 5 min. Slides were then immersed in 2xSSC for 5 min, followed by 95% and 100% ethanol for 5 min each at room temperature. For the mRNA amplification, primers designed to selectively bind tdTomato were used. 120 μl of the amplification reaction solution (60 μl Taq Master Mix (Apex Bioresearch, 42–138), 2 μl 1 mM Digoxigenin-11-dUTP (Roche, cat# 11093088910), 6 μl 1 mM forward and reverse primer, 57 μl water) was added, and 5 amplification cycles were run at 95 °C for 30 min, 58 °C for 30 min, and 72 °C for 1 min.

Slides were immersed in 2 × SSC at room temperature for 30 min. For detection of the signal, sections were incubated in buffer 1 (150 mM Tris, 100 mM/NaCl, pH 7.5) for 30 min and followed by a 2 h incubation with a 1:5000 diluted AP-coupled anti-DIG antibody (Roche, cat # 11093274910) in buffer 2 (0.5% Boehringer blocking reagent in buffer 1). Sections were washed in buffer 1 for 30 min and then incubated in a 1:50 diluted NBT/BCT (Roche, cat #11681451001) in buffer 3 (100 mM Tris 100 mM/NaCl, 50 mM MgCl₂, pH 7.5) for 2 h. The staining reaction was stopped with TE buffer (10 mM Tris, 1 mM EDTA, pH 8.0) for 15 min. The non-specific background was removed in 95% ethanol for 1 h, followed by rinsing in water to dissolve potential crystals. Finally, sections were stained in Nuclear Fast Red Counterstain (Vector, H-3403–500), underwent dehydration, and were mounted.

### In situ PCR for the detection of AAV DNA
Paraffin sections (5 μm) were dewaxed, and antigen retrieval was performed using 10 mM sodium citrate buffer at 100 °C for 10 min. Subsequently, slides were given a 0.02 M HCl for 10 min at room temperature. Slides were immersed in 1× PBS for 5 min, followed by a 10 μg/ml proteinase K digestion for 20 min at room temperature. For the amplification step, 120 μl of the amplification reaction solution (60 μl Taq Master Mix (Apex Bioresearch, 42–138), 2 μl 1 mM Digoxigenin-11-dUTP (Roche, 11093088910), 6 μl 1 mM forward and reverse primer, 56 μl water) was added, and 20 amplification cycles were run at 95 °C for 30 min, 58 °C for 30 min, and 72 °C for 1 min. Slides were immersed the slides in 2× SSC at room temperature for 30 min. For detection of the signal, sections were incubated in buffer 1 (150 mM Tris, 100 mM/NaCl, pH 7.5) for 30 min and followed by a 2 h incubation with a 1:5000 diluted AP-coupled anti-DIG antibody (Roche, cat# 11093274910) in buffer 2 (0.5% Boehringer blocking reagent in buffer 1). Sections were washed in buffer 1 for 30 min and then incubated in a 1:50 diluted NBT/BCT (Roche, cat # 11681451001) in buffer 3 (100 mM Tris 100 mM/NaCl, 50 mM MgCl2, pH 7.5) for 2 h. The staining reaction was stopped with TE buffer (10 mM Tris, 1 mM EDTA, pH 8.0) for 15 min. The non-specific background was removed in 95% ethanol for 1 h, followed by rinsing in water to dissolve potential crystals. Finally, sections were stained in Nuclear Fast Red Counterstain (Vector, H-3403–500), underwent dehydration, and were mounted.

Primers:

tdTomato_For: 5'-ACATGGCCGTCATCAAAGA-3'
tdTomato_Rev: 5'-CTTGTACGGCCTGTCCCATGC-3'

### Reporting summary
Further information on research design is available in the Nature Portfolio Reporting Summary linked to this article.

## Data availability
All data is available in the manuscript or the supplementary materials. Source data are provided in this paper. TIRFA mice will be made available through mouse repositories. Source data are provided in this paper.

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

## Acknowledgements

Finanical support by The Alice and Y.T. Chen Center for Genetics and Genomics (K.-D.B.), Avachrome Inc., National Institute of Diabetes and Digestive and Kidney Disease (DK134408 to K.D.B. and A.A.). National Institutes of Health (U01EB028901 to C.A.G.). H.L.S. was supported by a National Science Foundation Graduate Research Fellowship (DGF 2139754). Mark A. Kay for sharing the AAVNP59 capsid.

## Author contributions

K.D.B., M.B. and F.P.P. designed the experiments and wrote the manuscript. A.A. provided conceptual input, supervised AAV production and assisted with data analysis and writing the manuscript. M.B., S.C. and B.B.C., performed in vivo, T.C., T.J.G., H.S., Y.M. ex vivo experiments. M.B. and T.C. did IHC and quantifications. T.C. and S.H.O. did the DNA and mRNA in situ hybridization. T.J.G. and A.R. generated the AAV vectors. S.E.W., S.A.V., N.T.N.G., A.M.D. and J.G. assisted with human sample collection and preparation. H.S. and C.A.G. assisted with stem cell experiments.

## Competing interests

M.B., F.P.P., and K.D.B. are inventors on a patent application covering the presented technology. The remaining authors declare no competing interests.
