## [Peer Review File · Nature Communications]

Reviewers' Comments:

Reviewer #1:

Remarks to the Author:

In this manuscript Barzi et al combine two technologically advanced preclinical mouse models for in vivo human hepatocyte AAV transduction into a state-of-the-art system that holds great promise. The TIRFA model appears superior to current liver chimeric mouse models for testing AAV serotypes, particularly those that also transduce murine cells. Their use of AAV8 and AAV9 illustrates this point.

Major comments:

Starting with the seminal LK03 description (PMID: 24390344) the hepatocyte gene therapy field is moving beyond the mouse-hepatocyte tropic serotypes into ever more engineered human hepatocyte tropic variants (e.g. PMID 29055620). Therefore a comparison between AAV8 and a more recent human hepatocyte tropic serotype such as LK03 or NP59 would more clearly show the benefits and limitations of TIRFA mice over conventional TIRF/FRG or other human liver chimeric mouse models.

While the PHH data are convincing, the rationale nor findings for the teratoma and poorly defined pediatric liver cancer xenograft model can easily be understood. Which cell types in the teratoma were transduced and which not? And what percentage of the cancer xenograft expresses dTomato when engrafted in TIRF mice as opposed to TIRFA mice?

Minor comments:

Page 3 Abstract: This needs to be extensively rewritten. Some example:

Line 3: humanized is too vague.

Line 3: Why is 'transgene-free' relevant for this manuscript since the original FRG model is not being compared to TIRF model?

Line 6: 'other human tissues' is too vague, this comprises an iPSC teratoma and an undefined pediatric liver cancer xenograft and should be described specifically.

Page 3: 'While non-human primates are likely to remain most predictive for AAV gene therapies in humans, ethical and cost constraints limit these models.'

A recently reported model comparison PMID: 36927149 suggests NHP may only be more predictive for human hepatocyte transduction for certain serotypes. This statement should be modified.

Page 4 Line 2: 'Unfortunately, many AAV serotypes validated for transduction efficiency in humanized mice transduce murine hepatocytes much better than human hepatocytes^{14, 15}.' Why is the Paulk paper cited here? Fig S6 in that paper seems to contradict this manuscript.

Page 4 2nd paragraph: 'displaying increased concentration over time with increased humanization (Fig. 1d).' Figure 1d does not support this statement. hAlb kinetics over time should be presented or statement revised.

Page 4 3rd paragraph: what was the rationale for using such a high vector dose? Were other more typical AAV8 doses tested?

Page 5 3rd paragraph: Such a model should enable validation of AAV tropisms in any desired human tissue. Is there any literature that suggests teratoma layers predict terminally differentiated tissue AAV tropism or transduction efficiency? Please cite or adjust.

Page 5 3rd paragraph: 'For this purpose, we injected human induced pluripotent stem cells into TIRFA mice and let teratoma develop over 3-4 months.? If this was the experimental design why are the mice also containing human hepatocyte islands? Besides probably not accurately describing the experiment the rationale nor controls (TIRFA without PHH, TIRF with PHH and iPSC) are lacking here. This makes the data and advance of TIRFA mice over TIRF hard to interpret.

Page 6: 'known tissue markers': please cite GS and FAS as tissue markers and add conclusion

sentence to which layers are being transduced with AAV9-GFP.

Page 6: 'Similarly, we transplanted patient-derived human liver cancer tissue into humanized TIRFA livers and injected the same AAV vector and dose.' Is this an FLC, hepatoblastoma, other type of liver cancer? And it appears only few cells were transduced, so without this PDX engrafted in TIRF mice is hard to interpret whether this findings supports the underlying premise that TIRFA are superior.

Page 6: ' In summary' this paragraph is too vague and needs to be much more specific for what serotypes TIRFA mice are a huge advance to the field and for which questions their contributions may be more marginal.

Reviewer #2:

Remarks to the Author:

The manuscript A Humanized Mouse Model for Adeno-Associated Viral Gene Therapy by Barzi et al. generates a transgenic mouse model which knocks out AAVR in an immunodeficient mouse background such that they can be engrafted with human hepatocytes. The goal of the study was to make an improved model to identify AAV vectors with hepatocyte tropism that would likely translate to humans. The authors hypothesized that if they removed AAVR on mouse cells, the virus would no longer distribute to the mouse hepatocytes which function as a "sink" and would now be able to transduce human hepatocytes more efficiently. While the authors did demonstrate enhanced transduction of human hepatocytes in the new transgenic mice over the same background but with AAVR, there are major concerns which relate to both the conclusions of the findings as well as the overall usefulness of the model to better predict "translational" AAV capsids.

Comment 1: The biology of AAVR continues to be understood. Most data in the literature shows that it is needed for AAV transduction, but its role in uptake and AAV biodistribution is still unclear. Similarly in this study AAV transduction of the human hepatocytes over mouse hepatocytes is assessed, but whether the "sink" effect that the authors claim has been rectified, is unclear. In situ hybridization to detect AAV genomes in mouse vs human cells in the liver would need to be performed to confirm that AAVR deletion affects uptake in the liver, and not just transduction. Specifically, at a short time point after injection are there less AAV genomes in mouse vs human cells between the two mice (AAVR KO vs AAVR+/+)? If not, then the biodistribution and sink effect would still be occurring in the AAVR KO model (just the transduction is reduced in mouse cells).

Comment 2: Related to comment 1. While transduction of human hepatocytes clearly increases in the AAVR KO mice vs the AAVR containing mice, it is unclear the mechanism is due to that claimed. Could the increased transduction be due to longer vector half-life in blood because AAVR is KO on peripheral organs? This would be the case if AAVR affects biodistribution as the authors claim. The supplemental biodistribution data suggests that overall biodistribution is not affected, which brings back the question of comment 1: Is the sink effect really solved in this model?

Comment 3: If the point was to reduce transduction of murine cells in the liver (and not necessarily uptake) so that a higher signal to noise ratio can be achieved, then the text needs substantial revising.

Reviewer #3:

Remarks to the Author:

Barzi et al present a novel approach to test tropism of AAV vectors to human cells using a novel murine model. This is a very important advancement. Most viruses have species tropism. This has been a huge issue for preclinical assessment of efficacy of gene therapy vectors, and moreover for safety. Imagine testing wild type HIV on old world monkeys and concluding that this is safe for humans. Steins et al (PMC5033908) showed that an AAV evolved to infect pig airway epithelia failed to transduce human airways, similarly an AAV evolved to infect human airways failed to infect pig airways.

This model, in particular the liver component, should be able to be used to assess safety and efficacy of AAV vectors for gene therapy.

The conclusions are supported by the data, and the methodology is sound

Minor

1.- The statement 'while non-human primates are likely to remain most predictive...

Should have a reference or changed to: we don't even know if non-human primates is predictive.

RESPONSE TO REVIEWER COMMENTS

Reviewer #1:

In this manuscript Barzi et al combine two technologically advanced preclinical mouse models for in vivo human hepatocyte AAV transduction into a state-of-the-art system that holds great promise. The TIRFA model appears superior to current liver chimeric mouse models for testing AAV serotypes, particularly those that also transduce murine cells. Their use of AAV8 and AAV9 illustrates this point.

Major comments:

Starting with the seminal LK03 description (PMID: 24390344) the hepatocyte gene therapy field is moving beyond the mouse-hepatocyte tropic serotypes into ever more engineered human hepatocyte tropic variants (e.g. PMID 29055620). Therefore a comparison between AAV8 and a more recent human hepatocyte tropic serotype such as LK03 or NP59 would more clearly show the benefits and limitations of TIRFA mice over conventional TIRF/FRG or other human liver chimeric mouse models.

Our choice of AAV vectors to evaluate the TIRFA mouse was based on ongoing clinical trials and sustained interest of natural AAV serotypes. However, we agree with the reviewer that there are some novel and interesting recombinant capsids with increased tropism to human cells. Therefore, we evaluated the NP59 capsid in humanized TIRFA mice. As expected, the humanized TIRFA model is comparable to the humanized TIRF when using this capsid; murine cells are not transduced (a few cells in humanized TIRF) while there is comparable transduction efficiencies of human cells in both models. We discussed this constellation in a paragraph (discussion).

While the PHH data are convincing, the rationale nor findings for the teratoma and poorly defined pediatric liver cancer xenograft model can easily be understood. Which cell types in the teratoma were transduced and which not? And what percentage of the cancer xenograft expresses dTomato when engrafted in TIRF mice as opposed to TIRFA mice?

We clarified in the present version of the manuscript the rationale for the generation of teratoma and liver cancer in TIRFA mice. We think there is a good rationale to demonstrate utility of the TIRFA model for primary human cells and tissue other than the liver. The teratoma assay offers a model to address this question. However, we agree with the reviewer that our evaluation in teratoma (and also liver cancer) is limited. The reason being is that particularly the teratoma generates each time different primary tissue and even different proportions of the three germ layers. This extreme heterogeneity from one tumor to the other makes it very difficult to compare AAV transduction efficiencies. Therefore, we limited our evaluation to more descriptive co-stainings with tissue markers. However, we increased the panel of stainings in present version to respond to this criticism. Similarly, we have seen very heterogenous liver cancers since they are generated from different pieces of patient samples (primary xenografts). We described in a new paragraph the pros and cons of these experiments.

Minor comments:

Page 3 Abstract: This needs to be extensively rewritten. Some example:

Line 3: humanized is too vague.

Answer: Corrected to human liver chimeric

Line 3: Why is 'transgene-free' relevant for this manuscript since the original FRG model is not being compared to TIRF model?

Answer: The transgene-free is part of the strain name (see PMID: 34036256), which differs from the mouse referred as FRG mouse in regard to the modifications in the three genes and also the background strain. We don't know if the "transgene-free" per se matters for this work, but since the two mice strains are different and the work was done in TIRFA mice, we prefer to be accurate about the strain name.

Line 6: 'other human tissues' is too vague, this comprises an iPSC teratoma and an undefined pediatric liver cancer xenograft and should be described specifically.

Answer: We wrote a more specific abstract

Page 3: 'While non-human primates are likely to remain most predictive for AAV gene therapies in humans, ethical and cost constraints limit these models.'

A recently reported model comparison PMID: 36927149 suggests NHP may only be more predictive for human hepatocyte transduction for certain serotypes. This statement should be modified.

Answer: We modified this statement and added also this reference.

Page 4 Line 2: 'Unfortunately, many AAV serotypes validated for transduction efficiency in humanized mice transduce murine hepatocytes much better than human hepatocytes^{14, 15}.' Why is the Paulk paper cited here? Fig S6 in that paper seems to contradict this manuscript.

Answer: Thank you for catching this mistake, we corrected the references.

Page 4, 2nd paragraph displaying increased concentration over time with increased humanization (Fig. 1d). Figure 1d does not support this statement. hAlb kinetics over time should be presented or statement revised.

Answer: We added new data supporting our statement.

Page 4 3rd paragraph: what was the rationale for using such a high vector dose? Were other more typical AAV8 doses tested?

Answer: We tested a high vector dose to ensure that the AAV dose does not limit AAVR binding. The AAV dose chosen (1×10^{12} gc/mouse or 5×10^{13} gc/kg) is similar to the currently approved clinical dose of Roctavian (6×10^{13} gc/kg). Importantly, we wanted to ensure saturated uptake into mouse and human hepatocytes as this demonstrates how robust the TIRFA model can be. Despite the high dose, we observe markedly lower expression in murine hepatocytes. We did not test lower doses.

Page 5 3rd paragraph: Such a model should enable validation of AAV tropisms in any desired human tissue. Is there any literature that suggests teratoma layers predict terminally differentiated tissue AAV tropism or transduction efficiency? Please cite or adjust.

Answer: We could not find any literature on AAV transduction in teratoma. However, the medical literature is very clear in regard to teratoma having the capacity to form all kinds of differentiated tissues of humans. We explained better this rationale in current version in a separate paragraph in the

discussion section.

Page 5 3rd paragraph: 'For this purpose, we injected human induced pluripotent stem cells into TIRFA mice and let teratoma develop over 3-4 months.? If this was the experimental design why are the mice also containing human hepatocyte islands? Besides probably not accurately describing the experiment the rationale nor controls (TIRFA without PHH, TIRF with PHH and iPSC) are lacking here. This makes the data and advance of TIRFA mice over TIRF hard to interpret.

Answer: In the new manuscript version we explained better the rationale and experimental design.

Page 6: ' known tissue markers': please cite GS and FAS as tissue markers and add conclusion sentence to which layers are being transduced with AAV9-GFP.

Answer: We expanded this part of the work (more markers) and made conclusions in a new paragraph in the discussion.

Page 6: 'Similarly, we transplanted patient-derived human liver cancer tissue into humanized TIRFA livers and injected the same AAV vector and dose.' Is this an FLC, hepatoblastoma, other type of liver cancer? And it appears only few cells were transduced, so without this PDX engrafted in TIRF mice is hard to interpret whether this findings supports the underlying premise that TIRFA are superior.

Answer: We defined the human liver cancer used (hepatoblastoma) and gave a clearer interpretation in the discussion. In short, the intention of the liver cancer (and teratoma) work was to demonstrate feasibility of multiple human tissues in the TRIFA mouse. Comparison between different AAV serotypes or transduction efficiencies of different tissues is far beyond the scope of this second part. We hope though that these feasibility studies will motivate other groups to engage into such studies.

Page 6: ' In summary' this paragraph is too vague and needs to be much more specific for what serotypes TIRFA mice are a huge advance to the field and for which questions their contributions may be more marginal.

Answer: In most recent manuscript version, we have dedicated a new paragraph in the discussion section on AAV serotypes (AAV8, AA9 and AAVNP59) used in the TIRFA mouse.

Reviewer #2:

The manuscript A Humanized Mouse Model for Adeno-Associated Viral Gene Therapy by Barzi et al. generates a transgenic mouse model which knocks out AAVR in an immunodeficient mouse background such that they can be engrafted with human hepatocytes. The goal of the study was to make an improved model to identify AAV vectors with hepatocyte tropism that would likely translate to humans. The authors hypothesized that if they removed AAVR on mouse cells, the virus would no longer distribute to the mouse hepatocytes which function as a "sink" and would now be able to transduce human hepatocytes more efficiently. While the authors did demonstrate enhanced transduction of human hepatocytes in the new transgenic mice over the same background but with AAVR, there are major concerns which relate to both the conclusions of the findings as well as the overall usefulness of the model to better predict "translational" AAV capsids.

Comment 1: The biology of AAVR continues to be understood. Most data in the literature shows that it is needed for AAV transduction, but its role in uptake and AAV biodistribution is still unclear. Similarly in

this study AAV transduction of the human hepatocytes over mouse hepatocytes is assessed, but whether the “sink” effect that the authors claim has been rectified, is unclear. In situ hybridization to detect AAV genomes in mouse vs human cells in the liver would need to be performed to confirm that AAVR deletion affects uptake in the liver, and not just transduction. Specifically, at a short time point after injection are there less AAV genomes in mouse vs human cells between the two mice (AAVR KO vs AAVR+/+)? If not, then the biodistribution and sink effect would still be occurring in the AAVR KO model (just the transduction is reduced in mouse cells).

Answer: Thank you for this insightful question. We agree with the reviewer that this is an important mechanistic question. Therefore, we did DNA and RNA in situ hybridization (new Fig. 4). The results are very interesting and question the role of AAVR in mediating transduction. Our newly generated data support the notion that AAVR acts as an essential host factor and possibly intracellular receptor, but not for surface binding. The latter is likely mediated by glycan attachment factors and AAVR is critical after uptake for transduction. Hence, while some non-specific uptake might occur in mouse hepatocytes, no transgene expression is seen due to lack of AAVR. We included a paragraph in the discussion with our interpretation of results.

Comment 2: Related to comment 1. While transduction of human hepatocytes clearly increases in the AAVR KO mice vs the AAVR containing mice, it is unclear the mechanism is due to that claimed. Could the increased transduction be due to longer vector half-life in blood because AAVR is KO on peripheral organs? This would be the case if AAVR affects biodistribution as the authors claim. The supplemental biodistribution data suggests that overall biodistribution is not affected, which brings back the question of comment 1: Is the sink effect really solved in this model?

Answer: We agree with the reviewer that the overall biodistribution is not affected and apologize for not clearly pointing this out in previous version of the manuscript. As described above, using in situ hybridization we could gain some further mechanistic insight, which greatly improved the manuscript and our understanding of AAVR. We do observe more DNA (and RNA) only after 3 days in the liver of TIRF compared to TIRFA mouse. This may be due to other AAV clearance or degradation mechanisms when AAVR is absent in mouse hepatocytes; however we do not know if (in addition) the AAV half-life is altered and therefore toned down mechanistic interpretations.

Comment 3: If the point was to reduce transduction of murine cells in the liver (and not necessarily uptake) so that a higher signal to noise ratio can be achieved, then the text needs substantial revising.

Answer: Knowing now that transduction is affected, we have extensively revised the text.

Reviewer #3 (Remarks to the Author):

Barzi et al present a novel approach to test tropism of AAV vectors to human cells using a novel murine model. This is a very important advancement. Most viruses have species tropism. This has been a huge issue for preclinical assessment of efficacy of gene therapy vectors, and moreover for safety. Imagine testing wild type HIV on old world monkeys and concluding that it is safe for humans. Steins et al (PMC5033908) showed that AAV evolved to infect pig airway epithelia failed to transduce human airways, similarly an AAV evolved to infect human airways failed to infect pig airways.

This model, in particular the liver component, should be able to be used to assess safety and efficacy of AAV vectors for gene therapy.

The conclusions are supported by the data, and the methodology is sound

Answer: Thank you for pointing out the necessity of such models and sharing our enthusiasm for the TIRFA mouse.

Minor

1.- The statement 'while non-human primates are likely to remain most predictive...
Should have a reference or changed to: we don't even know if non-human primates is predictive.

Answer: We modified this sentence in response to reviewer #1 and #3

Reviewers' Comments:

Reviewer #1:

Remarks to the Author:

My concerns have been fully addressed. And the additional data in response to Reviewer 2 have made the manuscript even more interesting. Well done.

Reviewer #2:

Remarks to the Author:

The authors have greatly improved the manuscript during this revision. The revised text now matches the conclusions with the results. The additional studies with in situ hybridization has increased the knowledge of how AAV transduction is working in this mouse model is working as well as provided important additional information for how this receptor is functioning for AAV biology.

Overall, this model should be useful for studies to assess in vivo transduction of human hepatocytes without confounding expression coming from transduced murine hepatocytes.

Reviewer #3:

Remarks to the Author:

The updated version has improved the quality of the manuscript